# Can We Calibrate a Daily Time-Step Hydrological Model Using Monthly Time-Step Discharge Data?

**Soham Adla [1],\*** , **Shivam Tripathi [2]** and **Markus Disse [1]**

1   TUM Dept. of Civil, Geo and Environmental Engineering, Technical University of Munich,
    80333 München, Germany
2   Hydraulics and Water Resources Engineering, Dept. of Civil Engineering, Indian Institute of Technology
    Kanpur, Kanpur 208016, India
\*   Correspondence: soham.adla@tum.de

**Abstract:** Hydrological models are generally calibrated at longer time-steps (monthly, seasonal, or annual) than their computational time-step (daily), because of better calibration performance, lower computational requirements, and the lack of reliable temporally-fine observed discharge data (particularly in developing countries). The consequences of having different calibration and computation time-steps on model performance have not been extensively investigated. This study uses the Soil and Water Assessment Tool (SWAT) model to explore the correctness of calibrating a hydrological model at the monthly time-step even if the problem statement is suited to monthly modeling. Multiple SWAT models were set up for an agricultural watershed in the Indo-Gangetic basin. The models were calibrated with observed discharge data of different time-steps (daily and monthly) and were validated on data with the same or different time-steps. Intra- and inter-decadal comparisons were conducted to reinforce the results. The models calibrated on monthly data marginally outperformed the models calibrated on daily data when validated on monthly data, in terms of $P\text{-}factor$, $R\text{-}factor$, the coefficient of determination ($R^2$), and Nash–Sutcliffe Efficiency ($NSE$). However, the monthly-calibrated models performed poorly as compared to daily-calibrated models when validated on daily discharge data. Moreover, the daily simulations from the monthly-calibrated models were unrealistic. Analysis of the calibrated parameters revealed that the daily- and monthly-calibrated models differed significantly in terms of parameters governing channel and groundwater processes. Thus, though the monthly-calibrated model captures the patterns in monthly discharge data fairly well, it fails to characterize daily rainfall-runoff processes. The results challenge the existing practice of using different calibration and computation time-steps in hydrological modeling, and suggest that the two time-steps should be the same, irrespective of the time-step required for modeling.

**Keywords:** hydrological modeling; model calibration; model performance; model validation; SWAT model; rainfall-runoff; observation frequency; uncertainty

---

## 1. Introduction

Continuous hydrological models perform their computations (water balance) typically at the daily time-step. They represent hydrological processes by conceptual or physical parameterization. In general, the parameters of a hydrological model cannot be obtained from either field measurements or prior estimation, and hence, they are obtained from model calibration [1]. The literature review reveals that more often than not, the calibration is performed at a longer (monthly, seasonal, or annual) time-step than the computational time-step (daily). The consequences of having different calibration and computation time-steps on model performance have not been extensively investigated [2].

Hydrological modelers often resort to calibration at a longer time-step because: (i) model simulations are desired at longer time-steps (e.g., monthly or seasonal forecasts and climate change impact assessment); (ii) the output data for calibration are not available at daily time-steps (discharge data that are typically used for calibrating hydrological models may not be available at daily time-steps) [3–5]; (iii) the input data at daily time-steps may not be reliable (general circulation model rainfall simulations that drive hydrological models for climate impact assessment studies are considered to be less reliable at daily than at monthly or longer time-steps); (iv) the model performance during calibration is better at longer time-steps [6,7]; and (v) calibration at longer time-steps is computationally less intensive [2,8].

A case in point is the Soil and Water Assessment Tool (SWAT) model [9,10], which performs water balance at a daily time-step [11], but is generally calibrated monthly or annually [2,12]. At the time of writing (i.e., March 2018), an overview of 115 SWAT simulation modeling studies ADD HERE provided a performance-based understanding of different problem statements attempted using SWAT modeling in varied conditions. Out of the 114 studies that calibrated/validated the SWAT model, only 42 (37%) reported daily calibration-validation statistics. Another 14 (12%) reported only daily calibration statistics, without any validation statistics. This suggests that a large fraction of SWAT models is calibrated at time-steps coarser than the daily time-step. In particular, hydrological modeling studies for larger basins tend to be calibrated using monthly time-step observations [13,14]. The calibration at a daily time-step is even rarer for developing countries that have a scarcity of discharge data, like the transboundary Ganga River on the Indian subcontinent [15].

Lately, there has been an increase in daily SWAT hydrological simulations [16]. Nevertheless, it has been reported that daily prediction results are comparatively poorer than the monthly and annual predictions, notwithstanding a few exceptions, which are attributed to inadequate spatial representation of rainfall, errors in streamflow measurements, and various other issues [16,17]. A comparison of 22 SWAT applications (spanning five continents across a variety of problem statements) maintained that the 'strongest' reported results, based on the Nash-Sutcliffe Efficiency ($NSE$) and coefficient of determination ($R^2$) performance indices, are from annual and monthly model studies [18]. This is corroborated by previously-compiled studies as well [17–20].

Sudheer et al. [2] studied the effect of the calibration time-step on the performance of hydrological models. They proposed that the time scale of calibration could be critical in continuous time-step hydrological models like SWAT or SWM (Stanford Watershed Model [21]) and concluded that the model performance at a shorter time-step was not guaranteed by calibrating the model parameters at a longer time-step [2].

This study draws its motivation from Sudheer et al. [2] and addresses a complimentary problem statement. The objective is to explore the correctness of calibrating a hydrological model at the monthly time-step even if the problem statement is suited to monthly modeling. The hypothesis tested is as follows: given the availability of daily data for a monthly modeling study, it is mandatory that a model that performs computation at a daily time-step is calibrated at the daily time-step. The performances of the daily- and monthly-calibrated models are analyzed to study the differences in the representations of hydrological processes by the two models. The hypothesis is evaluated using the SWAT hydrological model with the Punpun River Basin (a tributary of River Ganga in Northern India) as a test case.

## 2. Materials and Methods

### 2.1. Study Area and Data Used

The Punpun River (Figure 1) originates in the hills of Palamau District (Jharkhand) around 24°11′ N, 84°9′ E [22], and joins the river Ganga at Fatuha, about 25 km downstream of Patna [23]. The delineated basin area of the Sripalpur gauging site as the basin outlet is 5495 km². The basin had no major hydraulic structure operational during the study period.

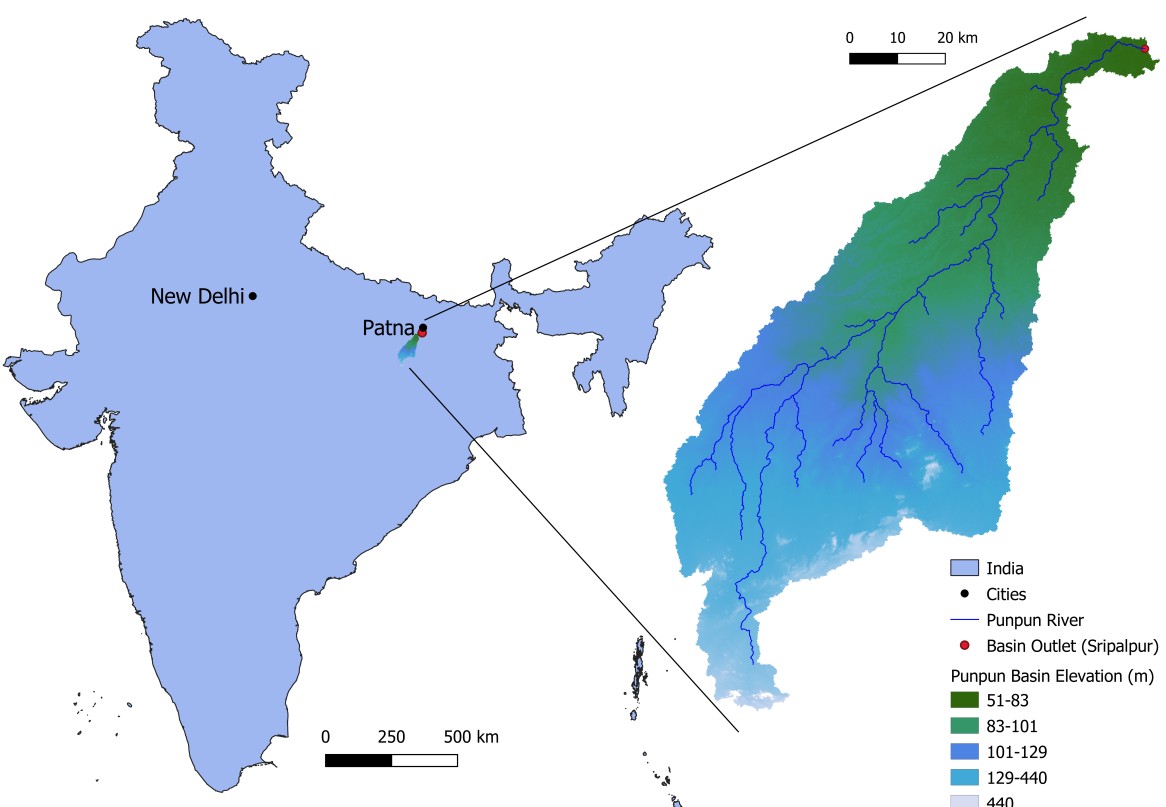

**Figure 1.** Study area (Punpun River Basin): Topographic map showing the Digital Elevation Map (DEM) with elevation in meters above mean sea level.

Parts of the river basin are inundated during the monsoon season (when it contributes a substantial amount of flow to the Ganga), but it also 'lives dry' during the low-flow season [22]. Consequently, the basin has been identified as a priority watershed for mitigative action against both flood and drought impacts [24,25]. Previous hydrological modeling studies in the Punpun River Basin have attempted rainfall-runoff modeling using artificial neural networks [22], Land Use and Land Cover (LULC) change detection using satellite imagery [23], and ecohydrological modeling using the SWAT model [26].

The setting up of the SWAT model requires topographic, land use, soil, weather, and management data. The input data used for the model setup are summarized in Table 1. The elevation in the basin ranges from 51–605 m Above Sea Level (ASL). The majority of Land Use (LU) is agricultural (68%), followed by fallow land (14%), and deciduous forests (4%). Soils of Hydrologic Group C/D (U.S. NRCS classification) cover most (81%) of the area, which indicates that they have low to very-low infiltration rates when thoroughly wetted and correspondingly high runoff potential [27]. As the study region has primarily agricultural LU, it was essential to incorporate local LU management practices to ensure representativeness. The LU management practices incorporated by Kaur et al. [28] for modeling the neighboring Damodar-Barakar Basin were adapted in the study. The discharge dataset used for calibrating the model was provided by the Central Water Commission (CWC) for its gauging site at Sripalpur (25°30′6″ N, 85°6′8″ E).

**Table 1.** Datasets used for the model setup.

| Data | Spatio-Temporal Resolution | Time-Period | Source |
|---|---|---|---|
| Discharge Data | Point data (Gauging site at Sripalpur 25°30′6″ N, 85°6′8″ E); daily | 1957–1999 | Central Water Commission (Ministry of Water Resources, River Development and Ganga Rejuvenation, Government of India) |
| Topographic Data (digital elevation model raster data) | 90 m × 90 m | 2000 | Shuttle Radar Tomography Mission (SRTM) |
| Land Use Land Cover (LULC) map | 1:250,000 scale | 2007–2008 | National Remote Sensing Centre (NRSC), Indian Space Research Organization (ISRO) |
| Soil Map | 1:250,000 scale | 2007–2008 | NRSC National Bureau of Soil Survey (NBSS) |
| Precipitation Data | 0.25° × 0.25°; daily | 1901–2015 | India Meteorological Department |
| Temperature | 1° × 1°; daily | 1948–2006 | Princeton University weather dataset |
| Relative Humidity | Modeled average data for the study region; daily | 1979–2014 | Global weather data for SWAT (NCEP Climate Forecast System Reanalysis) |
| Solar Radiation | 1° × 1°; daily | 1948–2006 | Princeton University weather dataset |
| Wind Speed | 1° × 1°; daily | 1948–2006 | Princeton University weather dataset |

*2.2. The SWAT Model*

The SWAT model was developed by the U.S. Department of Agriculture (USDA) to assist water resource managers in assessing impacts of land use management on large ungauged catchments [11]. Presently, the model is claimed to have the capabilities to address a wide variety of water resources and environmental problems due to its comprehensive nature, strong model support, and open-access source code [18].

The model itself is a result of about 40 years of research, originating in the models of the USDA Agricultural Research Service (ARS) like the Chemicals, Runoff and Erosion from Agricultural Management Systems (CREAMS) model [29], the Groundwater Loading Effects on Agricultural Management Systems (GLEAMS) model [30], and the Environmental Impact Policy Climate (EPIC) model [31,32]. Gassman et al. [18] expressed that the current model [10,17,33] is sustained by online documentation [27,34], multiple Geographic Information System (GIS) interfaces [35,36], and online resources [37].

The SWAT model framework divides the catchment into multiple sub-basins, each of which is further subdivided into Hydrologic Response Units (HRUs). An HRU is the smallest unit of computation in the SWAT model. It represents a unique combination of land use, soil, and slope class characteristics, but has no spatial reference (the sub-basin is the smallest unit with spatial reference). The SWAT model computes the daily water balance according to the following equation [34]:

$$SW_t = SW_0 + \sum_{i=1}^{t} (R_{day,i} - Q_{surf,i} - w_{seep,i} - E_{a,i} - Q_{gw,i}) \tag{1}$$

where:

| | |
|---|---|
| $t$ | = time period of simulation (days) |
| $SW_t$ | = final soil water content (mm) |
| $SW_0$ | = initial soil water content on day $i$ (mm) |
| $R_{day,i}$ | = amount of precipitation (including snowfall) on day $i$ (mm) |
| $Q_{surf,i}$ | = amount of surface runoff on day $i$ (mm) |
| $w_{seep,i}$ | = amount of water leaving the soil profile and entering the vadose zone on day $i$ (mm) |
| $E_{a,i}$ | = amount of evapotranspiration on day $i$ (mm) |
| $Q_{gw,i}$ | = amount of return flow on day $i$ (mm) |

A SWAT simulation can be separated into two phases: the soil/land phase of the hydrologic cycle in which the processes on the soil and sub-surface soil occur along with the circulation of nutrients, sediment and pesticides; and the routing phase of the hydrologic cycle, in which the circulation of water and sediment occurs through the river network to the outlet of the unit of computation [33].

Hydrological processes simulated by the SWAT model include canopy storage, surface runoff, infiltration, lateral subsurface flow, groundwater flow, redistribution of water within the soil profile, percolation, evapotranspiration, transmission losses, and pond recharge [33,38]. The surface runoff estimation, based on the CREAMS runoff model [29], includes the computation of runoff volume (based on either the SCS curve number method [39] or the Green and Ampt infiltration method [40]), peak runoff rate (based on the modified rational formula [41]), watershed time-of-concentration (estimated using Manning's formula accounting for overland and channel flows), and rainfall intensity during the watershed time-of-concentration [38]. The plant growth model within SWAT can estimate water and nutrient uptake from the root zone, transpiration, and bio-mass production [33]. SWAT estimates percolation and flow in each soil layer in the root zone using a storage routing technique, and lateral subsurface flow and recharge beyond the lowest soil layer are simultaneously calculated [38]. Percolation beyond the shallow aquifer contributes to the deep aquifer and is lost from the water balance computation [34].

Once the loadings of water, sediment, nutrients, and pesticides from the land phase to the channel are determined, they are routed through the streams and reservoirs [33]. The groundwater flow component is simulated by routing a shallow aquifer storage to the stream [42]. Potential ET can be calculated using the Hargreaves method [43], the Priestley–Taylor method [44], or the Penman–Monteith method [45]. More details of the model and its equations are provided in Arnold et al. [9] and Neitsch et al. [34]. In this study, Potential ET was calculated using the Penman-Monteith method, and the SCS curve number method was chosen for runoff partitioning.

*2.3. Sensitivity Analysis, Model Calibration, and Assessment Criteria*

Sensitivity analysis is a prerequisite to identify key parameters prior to model calibration [46]. It is the procedure to determine the rate of change in model output in response to changing model parameters [33]. The sensitivity analysis can be either local (changing one parameter at a time) or global (changing all parameters simultaneously) [47]. While local/One-At-a-Time (OAT) sensitivity analysis is limited because the sensitivity of one parameter depends on the values of the other parameters, global sensitivity analysis is challenging because of the large number of simulations required [33]. Herein, OAT sensitivity analysis was conducted using the Latin Hypercube (LH) sampling technique [48].

Model calibration was conducted using the Sequential Uncertainty Fitting procedure Version 2 (or SUFI2; [49,50]), which is a component of the SWAT-CUP (Calibration and Uncertainty Programs) software [47]. The procedure assumes that hydrological inverse problems are not uniquely solvable [50], and like in other Bayesian inverse methods, all parameter distributions producing the desired prediction uncertainty are potential solutions [49]. The SUFI2 procedure is similar to the Generalized Likelihood Uncertainty Estimation (GLUE) [51,52] in that the procedure is applied to parameter sets rather than individual values (to account for interactions between parameters explicitly), but different in that the key output is a set of 'best ranges' of all parameters, rather than a set of single-valued best parameters in GLUE [49].

SUFI2 consists of a sequence of steps to find a parameter distribution by iteratively optimizing an objective function to achieve a set of fitted parameter ranges that satisfy certain performance criteria [50]. In SUFI2, all the sources of uncertainty (input, conceptual, measurement, parameter, etc.) are expressed as input parameter uncertainty [47]. The input parameter uncertainties are depicted as uniform distributions, and LH sampling [53] is used in combination with a global search algorithm to achieve an improved objective function behavior [49]. The objective function chosen in this study was Nash–Sutcliffe Efficiency (*NSE*) [54], which is among the most widely-reported statistical tests in the literature [16,33].

The initially-chosen larger absolute intervals for parameters were iteratively updated (made smaller) around the 'best' performing set of parameter values, until the recommended thresholds of three performance indicators (details in Table 2): *P-factor*, *R-factor*, and the coefficient of determination, $R^2$, are met [49,50]. The model output uncertainty (not necessarily Gaussian) is depicted by the 95% Prediction Uncertainty (95PPU), which was computed at the 2.5% and 97.5% levels of the cumulative distribution of the output variable (here, basin outlet discharge) [50]. Readers are referred to Abbaspour et al. [49,50] for a detailed description of the SUFI2 procedure.

**Table 2.** Performance criteria followed in the SUFI2 procedure.

| Performance Index | Description | Recommended Value |
|---|---|---|
| *P-factor* | Fraction of the observed data (with its error) bracketed within the simulated 95% Prediction Uncertainty (95PPU) band [14]. The *P-factor* can lie between 0 and 1, and is ideally 1, indicating full capture of the hydrological processes by the model [33]. Model error can be given by 1-(*P-factor*) [14]. | For high-quality measurements, a *P-factor* > 0.8 is recommended; for low quality data, a *P-factor* > 0.5 is recommended [50] (used in this study); for discharge, a value of *P-factor* > 0.7 or 0.75 is recommended, depending on the project scale of input and calibration data adequacy [14]. |
| *R-factor* | The ratio of the mean width of the 95PPU band and the standard deviation of the observed data [14]. | An *R-factor* < 1.5 is recommended [49,50]. |
| $R^2$ (coefficient of determination) | $R^2$ measures the proportion of the variation in the measured data explained by the model [55] and is also defined as the squared value of the coefficient of correlation according to Bravais–Pearson [56]. $R^2$ can vary between 0 and 1; a higher value implies lesser variance [55]. | $R^2 > 0.5$ is recommended [57] |
| Nash–Sutcliffe Efficiency (*NSE*; used as the objective function) | Normalized measure determining the relative magnitude of the residual variance (analogous to 'noise') compared to the measured data variance (analogous to 'information') [54]. The range of *NSE* lies between 1.0 (perfect fit) and $-\infty$ (minus infinity). | *NSE* > 0.5 is recommended for hydrological simulations at the monthly time-step, with appropriate relaxation of the standard at the daily time step [55]. |

## 2.4. Model Setup and Modeling Experiments

The SWAT model's ArcGIS graphical user interface (ArcSWAT v.2009; [36,58] was used for pre-processing the input data. Subsequently, the freely-available SWAT 2012 (ver. 635) was used to run the model.

2.4.1. Calibration Cases

The SWAT models were independently calibrated for daily and monthly discharge data. To increase the confidence of the results, the calibration schemes were replicated across two time periods (each spanning roughly a decade): the 1980s (1979–1988) and the 1990s (1990–1997). This combination of the temporal resolution (daily or monthly) of the observed data and time periods of model simulation (1980s or 1990s) led to four calibration cases, which are summarized in Table 3.

**Table 3.** Calibration cases used in the study. Observed discharge data, measured at Sripalpur, were made available by the CWC.

| Nomenclature | Calibration Time-Step | Calibration Data Used | Calibration Period |
|---|---|---|---|
| D80 | Daily | Daily discharge | Warm-up period: 1979–1980; calibration period: 1981–1984 |
| M80 | Monthly | Monthly average of daily discharge | Warm-up period: 1979–1980; calibration period: 1981–1984 |
| D90 | Daily | Daily discharge | Warm-up period: 1990–1991; calibration period: 1992–1994 |
| M90 | Monthly | Monthly average of daily discharge | Warm-up period: 1990–1991; calibration period: 1992–1994 |

Additionally, each simulation time period was divided into calibration and validation sets: the 1980s model was calibrated using data from 1981–1984, and the 1990s model was calibrated using data from 1992–1994. The remaining periods (1985–1988, 1995–1997) were used as the validation periods for the different models. The respective models were developed with 2 warm-up years each (1979–1980 for the 1980s, 1990–1991 for the 1990s, respectively).

2.4.2. Model Calibration

One-At-a-Time (OAT) sensitivity analysis was conducted on parameters identified as contributors to the major SWAT model processes using the LH-OAT method [48]. These processes included groundwater, soil, runoff, channel, geomorphological, and evaporation components. Each of the 26 thus chosen parameters was tested for sensitivity for each calibration case.

The SWAT models developed for the Punpun Basin had 89 sub-basins with 1709 HRUs. Estimating or calibrating HRU-specific parameters for basins with many HRUs and limited data records is a modeling challenge [47]. Alternatively, each parameter can be lumped and subsequently calibrated by using a single global modification term with a corresponding prefix in the SWAT-CUP software (which scales initial estimates by a multiplicative 'r', additive 'a' , or replacement 'v' factor) [47]. Table 4 mentions the spatial scale of lumping of the identified sensitive parameters, the modification technique used in their calibration, and their default values (interval) determined using LU and soil input data.

Table 4. Calibrated SWAT parameters, spatial scale of variation, default values, method of modification during calibration, and relevant calibration cases.

| Sensitive Parameter Identified by LH-OAT | Definition of SWAT Parameters (Relevant SWAT File) [34] | Spatial Scale of Variation | Default SWAT Model Values (units) | Method of Modification: Additive 'a', Multiplicative 'r', Replacement 'v' (Initial Modification Range) | Calibration Cases (From Table 3) in Which the Parameter Was Calibrated |
|---|---|---|---|---|---|
| CH_K2 | Effective hydraulic conductivity in main channel alluvium (Routing file of each reach ' *.rte') | Sub-basin | 0 (mm/h) | v (0, 150) | D80, M80, D90, M90 (all) |
| GW_DELAY | Time taken in days for water to move from lowest soil layer to the first shallow aquifer (groundwater file of each HRU '*.gw') | HRU | 31 (days) | v (0, 450) | All |
| SOL_AWC | Available water capacity in the soil layer (soil file of each HRU '*.sol') | HRU | HRU dependent, different values (function of soil data) (mm $H_2O$/mm soil) | r (−0.5, 0.5) | All |
| GW_REVAP | Groundwater evaporation coefficient (groundwater file of each HRU '*.gw') | HRU | 0.02 (unitless) | v (0.02, 0.2) | All |
| ESCO | Soil evaporation compensation factor (1 file for the entire basin 'basin.bsn') | Basin | 0.95 (unitless) | v (0, 1) | All |
| CN2 | Initial SCS runoff curve number for Moisture Condition II (management file of each HRU '*.mgt') | HRU | HRU dependent, different values (60–84) (unitless) | r (−0.2, 0.2) | All |
| BASEFLOW ALPHA-FACTOR (ALPHA_BF) | Baseflow recession constant. Index for groundwater flow response to changes in recharge (groundwater file of each HRU '*.gw') | HRU | 0.048 (unitless) | v (0, 1) | D80, D90 |
| SURLAG | Surface runoff lag coefficient (1 file for entire basin 'basin.bsn') | Basin | 4 (unitless) | v (0.05, 10) | D80, D90 |

The parameters 'ESCO' and 'SURLAG' were calibrated at the basin scale. While SURLAG is defined as a single basin-wide value, SWAT allows ESCO to be modified at both the basin, as well as HRU scale [27]. The parameter 'CH_K2' is sub-basin specific, but was calibrated at the basin level because measured discharge data only at a single site were available for calibration, rather than data for each sub-basin outlet, which would be appropriate for sub-basin level calibration. The other parameters, 'SOL_AWC', 'CN2', 'GW_DELAY', 'ALPHA_BF', and 'GW_REVAP', were all calibrated at the HRU scale. Out of these, the parameters 'CN2' and 'Sol_AWC' were modified by a global modification factor (prefix 'r'), while the other 5 parameters were replaced (prefix 'v') after each calibration run.

The parameters identified after LH-OAT sensitivity analysis were subjected to the SUFI2 calibration procedure. This iterative procedure starts by randomly generating parameter sets from a pre-defined initial parameter range (Table 4). The model simulation was then performed for each parameter set. The number of simulations per iteration was set to 500 so as to strike a balance between the recommended number of simulations (300–1000 [14]) and available computational resources. The number of iterations required for convergence depends on the calibration performance (as described in Section 2.3) and is usually between 3 and 5 [14]. The 500 random samples from the parameter set of the last iteration were used in the validation runs.

### 2.4.3. Model Validation

The calibrated models for each calibration cases were validated on the following three independent datasets:

1.  **Self-validation using observed data of the same time-step and decade:** This follows the split-sample approach [59], requiring the use of sequential years within each period. Self-validation helped to assess the performance of the calibrated models and was performed for both the decades and both the modeling time-steps.
2.  **Validation using observed data of different time-steps, but the same decade:** Parameters for a calibration time-step were validated on a model setup using observed data from a different time-step (for example, daily-calibrated parameters of the 1980s model validated on the monthly 1980s data, represented as 'D80_v_M80'). This validation was conducted to test the central hypothesis of the study, i.e., whether a study requiring monthly simulation modeling is better attempted by a model calibrated using monthly or daily observed data (given the availability of daily data).
3.  **Validation using observed data of the same time-step, but different decades:** Model parameters for each calibration time-step were validated on the data of the same time-step, but from the other decade. This was conducted to further corroborate the validation results and check the decadal changes in the river basin characteristics from a hydrological modeling perspective.

## 3. Results and Discussion

### 3.1. Sensitivity Analysis

Table 4 lists the parameters selected by the LH-OAT sensitivity analysis from a restricted set of model parameters that capture the major processes in SWAT [48]. Six parameters were identified as sensitive to the predicted runoff for all four calibration cases (D80, D90, M80, and M90). These included two HRU-scale groundwater-related parameters (GW_DELAY and GW_REVAP coefficient), an HRU-scale management parameter for rainfall partitioning into runoff (CN2), an HRU-scale soil parameter determining available water capacity (SOL_AWC), a routing parameter representing effective hydraulic conductivity in the river channel (CH_K2), and a basin-scale parameter determining the depth distribution of the soil evaporation compensation (ESCO). The parameters SURLAG (basin scale) and the BASEFLOW ALPHA-FACTOR (HRU scale) that contribute to the surface and

groundwater response to runoff, respectively, were identified as sensitive only for the daily calibration cases (D80 and D90).

## 3.2. Performance of the SWAT Models

The model performance was evaluated using four measures, namely $NSE$, $R^2$, $P$-$factor$, and $R$-$factor$ (Table 2). Both $R^2$ and $NSE$ are based on the 'best simulation' (which corresponds to the highest objective function value across all simulations within the SUFI2 iteration) and do not account for simulation uncertainties, which are represented by the $P$- and $R$-$factors$.

### 3.2.1. Calibration Performance

In all four calibrations runs (D80, M80, D90, and M90), the $R^2$ and $NSE$ were satisfactory (Table 5). Within the same decade, the $R^2$ and $NSE$ were both relatively higher in the monthly calibration runs than the daily calibration runs. The lumping of daily discharge measurements (to monthly averaged values) compensated for positive and negative simulation errors, thereby improving the $R^2$ and $NSE$ values for monthly models. Comparing across decades, the models calibrated on 1990s data performed slightly better than those calibrated on 1980s data, and this difference was more pronounced in the monthly calibration runs. Within the same decade, the $P$- and $R$-$factors$ for monthly calibration runs were substantially better compared to their daily counterparts, for both decades. This suggests that the daily-calibrated model 95PPU band of simulated discharge was not able to capture a larger fraction of observations in comparison to the monthly-calibrated model 95PPU band. The $P$-$factor$ for both daily- and monthly-calibrated parameters increased from the 1980s to the 1990s, which partly reinforced the fact that the input data (topography, LULC, and soil maps) were more reflective of the 1990s than the 1980s (see Table 1).

**Table 5.** Model calibration and validation performance for all calibration cases. 'D80_c' implies the calibration of the model using daily data from the 1980s decade, and 'D80_v_M90' implies the validation of the model calibrated using daily 1980s data on the 1990s monthly data. Years in parenthesis indicate the time period used in the particular calibration or validation.

| | **1980s Daily** | | |
|---|---|---|---|
| | Calibration | Validation Cases | | |
| | | Self-validation | Different time-step, same decade | Same time-step, different decade |
| | D80_c (1981–1984) | D80_v_D80 (1985–1988) | D80_v_M80 (1985–1988) | D80_v_D90 |
| $P$-$factor$ | 0.22 | 0.23 | 0.25 | 0.31 |
| $R$-$factor$ | 0.37 | 0.46 | 0.35 | 0.63 |
| $R^2$ | 0.67 | 0.68 | 0.87 | 0.68 |
| $NSE$ | 0.66 | 0.66 | 0.80 | 0.68 |
| | **1990s Daily** | | |
| | Calibration | Validation Cases | | |
| | | Self-validation | Different time-step, same decade | Same time-step, different decade |
| | D90_c (1992–1994) | D90_v_D90 (1995–1997) | D90_v_M90 (1995–1997) | D90_v_D80 |
| $P$-$factor$ | 0.53 | 0.45 | 0.42 | 0.37 |
| $R$-$factor$ | 0.66 | 0.72 | 0.42 | 0.73 |
| $R^2$ | 0.67 | 0.70 | 0.86 | 0.73 |
| $NSE$ | 0.67 | 0.69 | 0.86 | 0.67 |

**Table 5.** *Cont.*

| | Calibration | Validation Cases | | |
|---|---|---|---|---|
| **1980s Monthly** | | | | |
| | | Self-validation | Different time-step, same decade | Same time-step, different decade |
| | M80_c (1981–1984) | M80_v_M80 (1985–1988) | M80_v_D80 (1985–1988) | M80_v_M90 |
| *P-factor* | 0.76 | 0.79 | 0.80 | 0.83 |
| *R-factor* | 0.52 | 0.56 | 0.76 | 0.55 |
| $R^2$ | 0.87 | 0.82 | 0.21 | 0.89 |
| *NSE* | 0.87 | 0.79 | −0.65 | 0.89 |
| **1990s Monthly** | | | | |
| | Calibration | Validation Cases | | |
| | | Self-validation | Different time-step, same decade | Same time-step, different decade |
| | M90_c (1992–1994) | M90_v_M90 (1995–1997) | M90_v_D90 (1995–1997) | M90_v_M80 |
| *P-factor* | 0.88 | 0.86 | 0.80 | 0.81 |
| *R-factor* | 0.52 | 0.63 | 0.57 | 0.63 |
| $R^2$ | 0.93 | 0.90 | 0.19 | 0.83 |
| *NSE* | 0.93 | 0.90 | −0.51 | 0.78 |

### 3.2.2. Validation Performance

**Self-validation**

The calibrated models were first validated with the independent observed data of the same-time step available within the same decade. For both time-steps (daily and monthly), the validation during the same decade (1985–1988 or 1995–1997) yielded $R^2$ and *NSE* values that were similar to those of the calibration period, indicating that the models performed equally well in both calibration and validation runs within the same decade. Further, similar to the results of the calibration, the monthly models had higher $R^2$ and *NSE* values than their corresponding daily models. Comparing across decades, it can be seen that the models had higher $R^2$ and *NSE* in the validation period of the 1990s compared to the 1980s, and this difference was more pronounced in the monthly calibration runs.

The *P*- and *R-factors* were also very similar during the calibration and self-validation runs, suggesting that the models were not overfitted. The monthly *P-factors* performed better than the daily *P-factors*, indicating that more observations fell into the 95PPU bands of the simulated outputs in the monthly self-validation runs than in the daily self-validation runs. The *R-factor* was higher for M80 (*R-factor* = 0.56) than D80 (*R-factor* = 0.46), but in the 1990s, M90 (*R-factor* = 0.63) had a slightly lower *R-factor* than D90 (*R-factor* = 0.72). Nevertheless, since all the *R-factor* values were less than 1.5, all the simulations were acceptable [14]. Comparing the self-validation performance across decades, the *P-factor* did not change substantially in the monthly runs, though in the daily runs, it almost doubled in the 1990s (D80 *P-factor* = 0.23, D90 *P-factor* = 0.45).

**Validation using observed data of different time-steps, but within the same decade**

A large difference in the performance was observed when the models calibrated using one time-step were validated on the other time-step, i.e., daily-calibrated model parameters validated on monthly data and vice versa. On the one hand, the performance measures of the daily-calibrated models improved when they were validated on the monthly data (for example, D80_v_D80: $R^2$ = 0.68, *NSE* = 0.66, but D80_v_M80: $R^2$ = 0.87, *NSE* = 0.80). This improvement was expected because the time averaging compensated for positive and negative errors in the daily simulations. On the other

hand, the monthly-calibrated models performed significantly more poorly on daily data (for example, M80_v_M80: $R^2 = 0.82$, $NSE = 0.79$, but M80_v_D80: $R^2 = 0.21$, $NSE = -0.65$). This raises questions on the ability of the monthly-calibrated model parameters to characterize the river basin, since any parameter set that claims to have a particular extent of representativeness should ideally lead to an equally good or bad performance irrespective of the time-step of the model run. This finding may have been overlooked if only monthly calibration-validation runs would have been performed.

The validation performance of the daily-calibrated models on the monthly data did not change significantly from one decade to another (i.e., D80 validated on M80 data and D90 validated on M90 data), but the M80 parameters validated on D80 data performed worse than the M90 parameters validated on D90 data. Particularly in the case of $NSE$, the M80 parameters had an $NSE = -0.65$ when validated on D80 data, and the M90 parameters had an $NSE = -0.51$ when validated on D90 data. The negative value of $NSE$ indicated that even the mean observation values had a better fit to the daily observed data than the monthly-calibrated SWAT models.

The daily-calibrated parameters validated on the monthly data yielded a lower *P-factor* (D80_v_M80: 0.25, D90_v_M90: 0.42) and a lower *R-factor* (D80_v_M80: 0.35, D90_v_M90: 0.42) than the corresponding indicators when the monthly-calibrated parameters were validated on the daily data (M80_v_D80 *P-factor*: 0.80, *R-factor*: 0.76; M90_v_D90 *P-factor*: 0.80, *R-factor*: 0.57). This implies that though the 95PPU bands of the monthly parameters better captured the daily observations, their width was wider when compared to the standard deviation of the observations, although they were all within acceptable limits [49,50].

**Validation using observed data of the same time-step, but different decades**

This analysis had two objectives. The first objective was to reinforce the results from each decade presented in the previous section. The other objective of assessing the decadal changes in the Punpun Basin from a hydrological modeling perspective was not central to the study and is presented briefly for completeness (for details, see Adla and Tripathi [60]).

Comparing the models across decades (without changing the calibration time-step), one can see that the D80 model performed roughly as well on the D90 data ($R^2 = 0.68$, $NSE = 0.68$) as the D90 model performed while being validated on the D80 data ($R^2 = 0.73$, $NSE = 0.67$). In the corresponding monthly result, the M80 model performed better when validated on the M90 data ($R^2 = 0.89$, $NSE = 0.89$) than when the M90 model was validated on the M80 data ($R^2 = 0.83$, $NSE = 0.78$).

Comparing the 'self-validation' runs of one decade to the validation runs of the models calibrated for the other decade (keeping the time-step common), the models calibrated for the 1990s performed equally or better than the models calibrated for the 1980s. The improvement was more substantial in the monthly runs as compared to the daily runs.

The ability of the 1980s parameters to simulate the 1990s data of the same time-step and the 1990s parameters' ability to simulate the 1980s data of the same time-step (w.r.t. the *P-factor* and *R-factor*) was higher for the monthly case than the daily case. This implied that the models that were parameterized and calibrated using lumped information were able to envelope the other decade's lumped observations more effectively than the models that were parameterized and calibrated using discretized (daily) observations. Overall, both the daily and monthly models performed equally well within the same decade as they performed in predicting the other decade. These results can only suggest that the models had some predictive ability across decades for the time-steps in which they were calibrated.

3.2.3. Calibrated Parameter Set

Figure 2a–h illustrates the ranges of the calibrated parameters of all the identified sensitive parameters. In each sub-plot, the calibrated parameter range is compared with its respective default values, for the corresponding calibration cases. The results of the global sensitivity analyses corresponding to all calibration and validation cases (based on Table 3 and Section 2.4.3) are given

in Table A1 in Appendix A. To discuss the effect of calibration time-step on the calibrated values, the parameters are divided into the following three groups:

1. **Parameters with differences across the calibration time-steps:** The effective channel hydraulic conductivity 'CH_K2' and groundwater delay time 'GW_DELAY' (Figure 2a,b) showed visible differences between daily- and monthly-calibrated parameter ranges. The daily-calibrated CH_K2 values were higher than the monthly-calibrated values, which implied that exchange of water through the channel bed was modeled as being faster in the daily-calibrated parameter set (the bed material corresponded to 'moderately high' to 'very high' loss/gain rates) than the monthly-calibrated parameter set ('insignificant' to 'moderate' loss/gain rates) [27]. This implied that the relatively higher variation in daily observations at the basin outlet was interpreted by the daily-calibrated model as coarser channel bed material, which facilitated relatively faster exchange of water between the channel and the groundwater layer.

   The daily-calibrated GW_DELAY values were markedly higher than the monthly-calibrated values, across both decades. This implied that the time taken by the water to percolate beyond the soil profile before recharging the shallow aquifer was estimated to be higher by the daily-calibrated model parameter set (median value greater than 150 days), as compared to the monthly-calibrated GW_DELAY values (median values less than 100 days) [27]. The GW_DELAY values estimated using the monthly data subsequently resulted in the rather irregular and unrealistic simulation of the daily streamflow (see Figure 3b).

2. **Parameters with differences across the decades:** Parameters that did not exhibit variation between daily and monthly calibration time-steps, but some variation across decades, included ALPHA_BF, SURLAG, GW_REVAP, and to a lesser extent, CN2.

   The parameters ALPHA_BF (Figure 2c) and SURLAG (Figure 2d), which were sensitive only for daily calibrations, had relatively higher ranges for the 1990s models than the 1980s models. An increase in ALPHA_BF implied a more rapid response of the baseflow to rainfall, i.e., a quicker decline of baseflow recession was seen in the 1990s compared to the 1980s. An increase in Surlag implied an increase in the fraction of surface runoff allowed to reach the respective reach on the same day (for a particular time of concentration). Hence, the 1990s saw a higher fraction of surface runoff storage reaching the model reach within a day, which was indicative of increased urban settlements in the basin [61].

   The central tendency of GW_REVAP (Figure 2e) was relatively lower in the 1990s in both daily and monthly calibrations, which implied that the rate of movement of water from the shallow aquifer to the root zone was simulated to be higher (as a response to the evapotranspiration demand) in the 1980s than the 1990s (where this upward movement was restricted). The types of plants in the basin, particularly deep-rooted plants (found more in forests) also influenced the GW_REVAP coefficient, and hence, this decrease in the tendency for 'revap' can be potentially due to the decrease in forest cover from the 1980s to the 1990s [61].

   The initial SCS runoff curve number (CN2, Figure 2f) for average soil moisture conditions did not show clear differences in the calibrated values. For the daily model, the CN2 increased from the 1980s to the 1990s, which indicated higher impermeability of the soil and, hence, more fractionation of rainfall into surface runoff generation.

3. **Parameters with no discernible variation:** The soil evaporation compensation coefficient 'ESCO' (Figure 2g) and available water capacity 'SOL_AWC' (Figure 2h) did not show any apparent differences between calibration time-steps and across decades. The decrease in ESCO across all calibration cases from the default value implied that the model allowed more evapotranspirative demand to be met through water stored in the deeper soil layers. The ranges of SOL_AWC varied between 0.03 and 0.21 without any clear differences across calibration cases, implying that the soil water storage was conceptualized similarly by the SWAT models calibrated with both the daily and monthly observations and across both decades.

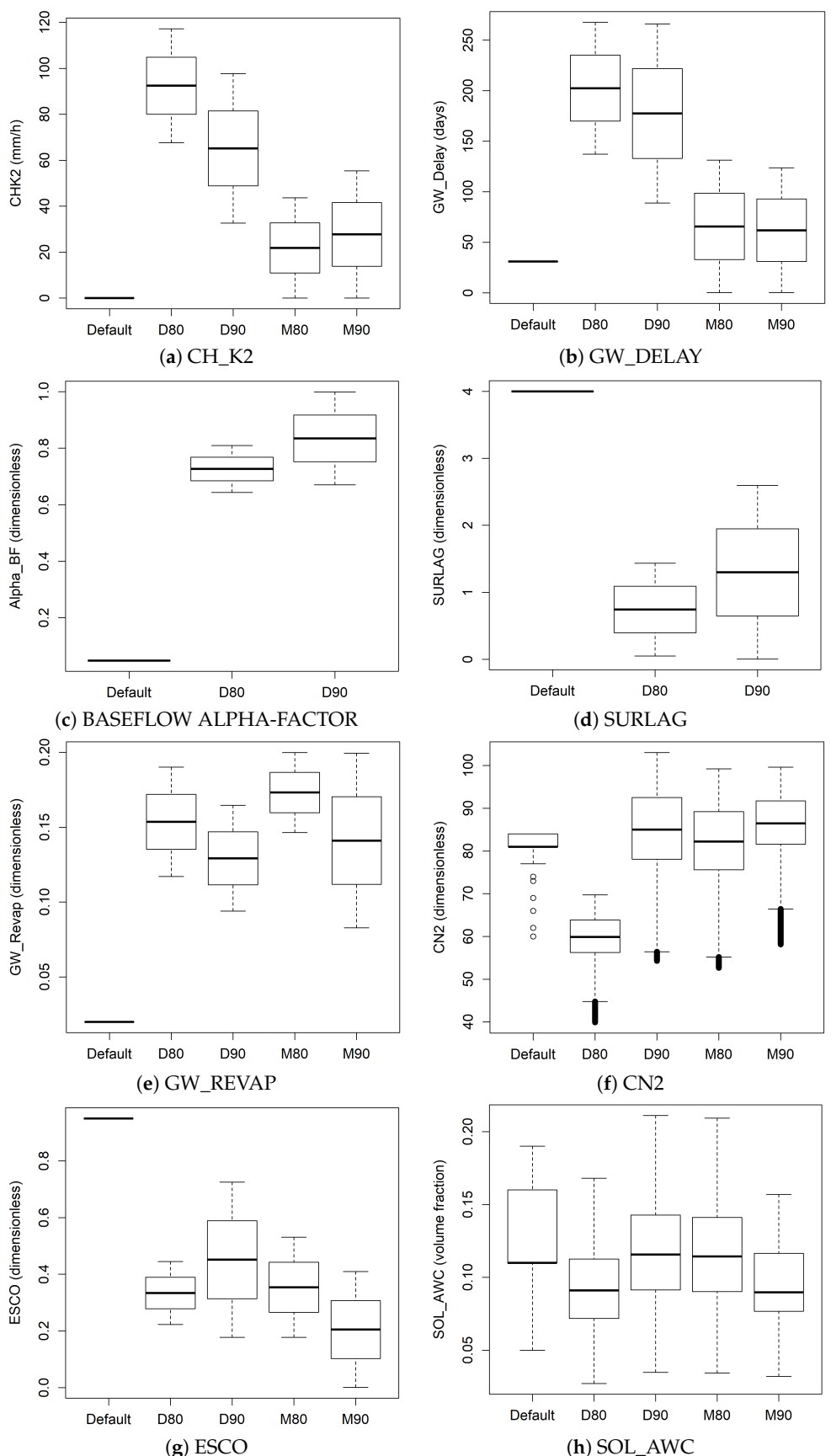

**Figure 2.** Calibrated parameter set of basin lumped parameters. Boxplots show parameter ranges. (**a**) CH_K2 and (**b**) GW_DELAY had differences across calibration time-steps; (**c**–**f**) ALPHA_BF, SURLAG, GW_REVAP, and CN2 showed (variable extent) decadal changes; and (**g**) ESCO and (**h**) SOL_AWC did not show any discernible trends.

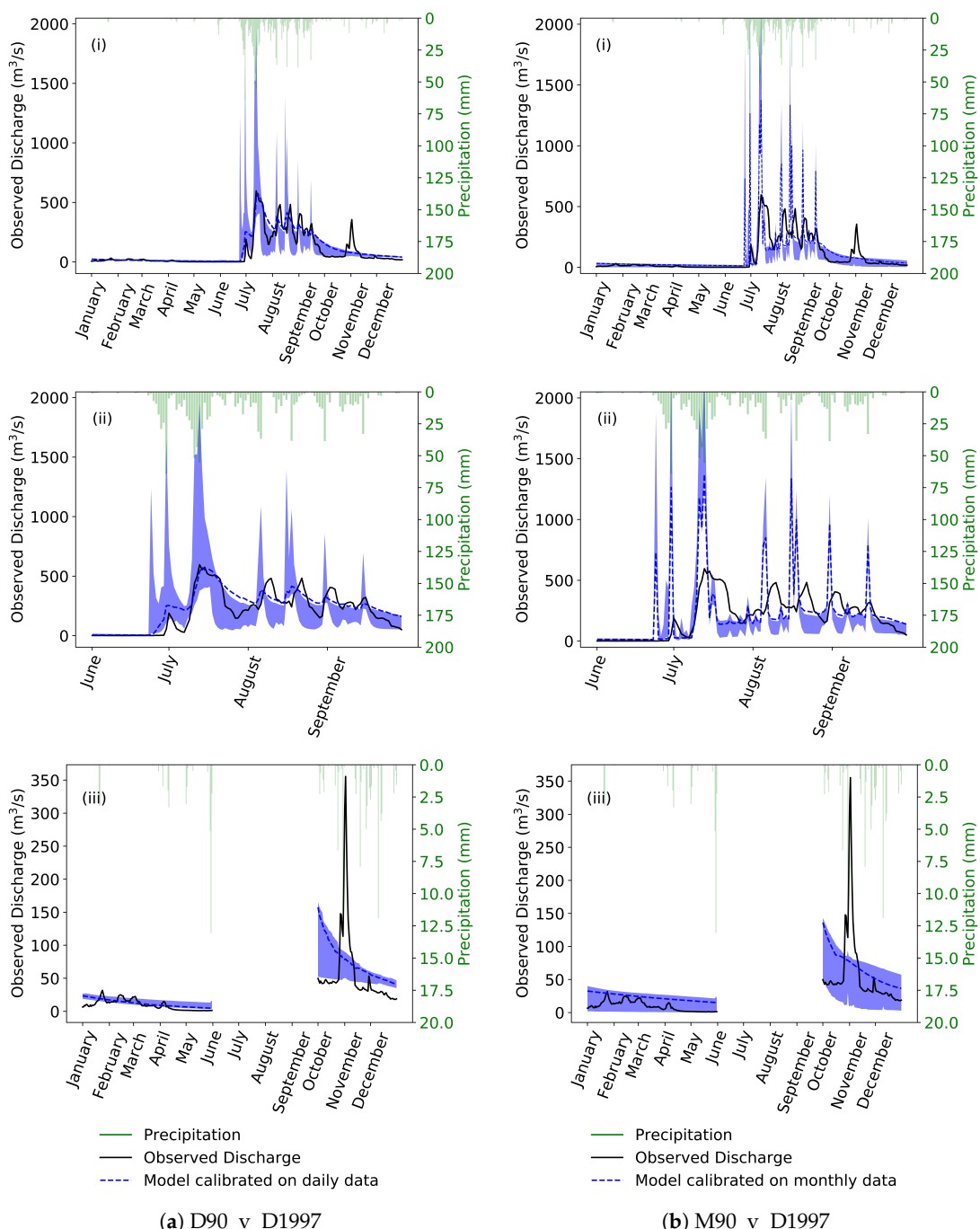

**Figure 3.** Time-series comparison of SWAT simulated and observed basin outlet daily discharges for the year 1997, developed using calibrated parameter sets of the calibration cases (**a**) D90 (D90_v_D1997) and (**b**) M90 (M90_v_D1997), illustrated for different periods: (i) full year, (ii) monsoon (June–September), and (iii) non-monsoon (data only presented for January–August and October–December). The blue band indicates the 95PPU range of simulated discharge; the dotted line indicates the simulation corresponding to the 'best' parameter set; and observations are represented by the solid line.

### 3.2.4. Streamflow Simulations

The validation results of the SWAT models discussed in the preceding sections did not justify the use of monthly-calibrated parameters, despite the inferior calibration performance of the daily-calibrated SWAT models. Nevertheless, it was felt that a comparative assessment of the

performance indices and resultant representative parameters would need a supplementary analysis of the simulated and observed discharge time series. This analysis was carried out to investigate closely the rainfall-runoff responses of the models calibrated at different time-steps.

The 1990s decade was better modeled by both monthly and daily SWAT models. Hence, a specific year within the 1990s decade, calendar year 1997, was selected (for illustration purposes; results are given in Figures 3 and 4), for comparing streamflow simulations at daily and monthly time-steps.

1.  Daily simulations (Figure 3):

    The mean discharges of the 'best simulation' from the models calibrated at daily (104.0 cumecs) and monthly (103.7 cumecs) time-steps were similar, but higher than the mean of the observed daily discharge (87.20 cumecs). The standard deviation of the observed daily discharge (133.03 cumecs) was closer to the standard deviation of the 'best simulation' from the daily-calibrated model (133.4 cumecs) than the monthly-calibrated model (186.56 cumecs).

    The hydrographs simulated by the monthly-calibrated model had steep rising and falling limbs, with peaks significantly larger than the observed peaks. These unrealistic peaks could be attributed to the relatively small 'GW_DELAY' parameter. The falling limb of the daily-calibrated model was relatively flat compared to the observations. However, the flood peaks simulated by the daily-calibrated model were comparable to the observations, and overall, the simulation looked more realistic.

    The 95PPU band was wider for the daily-calibrated model (average width: 100.9 cumecs) compared to the monthly model (average width: 83.9 cumecs). Surprisingly, the *P-factor* of the monthly-calibrated model (0.79) was higher than the daily-calibrated model (0.42). This apparent anomaly can be explained by separately analyzing monsoon (June–September, which received 89% of the annual rainfall) and non-monsoon periods. In the monsoon period, the daily-calibrated model (*P-factor*: 0.52) performed better than the monthly model (*P-factor*: 0.50), but during the non-monsoon period, the monthly model (*P-factor*: 0.93) greatly outperformed the daily model (*P-factor*: 0.36).

    It is observed that both the daily- and monthly-calibrated models, with their best simulations and respective 95PPU bands, were unable to capture the large discharge in the hydrograph from 26 October to 12 November 1997. Such peaks are observed during the winter months of other years as well, and may occur through a combination of winter rainfall and irrigation application during the cropping season. However, a more detailed on-site investigation is required for a better understanding of this phenomenon.

2.  Monthly simulations (Figure 4):

    The mean and standard deviation of the observed monthly data were 86.54 cumecs and 115.96 cumecs, respectively. These values were closer to the corresponding statistics of the 'best simulation' from the daily-calibrated model (mean: 97.3 cumecs and standard deviation: 128.59 cumecs) than the monthly-calibrated model (mean: 104.6 cumecs and standard deviation: 134.08 cumecs), though both the models overpredicted discharge. Taking into account the other performance statistics ($R^2$-daily: 0.87, monthly: 0.95; $NSE$-daily: 0.83, monthly: 0.89), it can be concluded that the monthly-calibrated model marginally outperformed the daily-calibrated model in simulating monthly average discharge for a single year.

    The 95PPU band was larger (average width: 78.1 cumecs) and slightly more variable (standard deviation of width: 57.1 cumecs) for the monthly-calibrated model compared to the daily model (average width: 47.5, standard deviation of width: 49.4). Understandably, the *P-factor* of the monthly-calibrated model (0.92) was significantly higher than the daily-calibrated model (0.50). The daily-calibrated model kept a similar level of performance in both monsoon and non-monsoon periods (both *P-factors*: 0.5). However, the *P-factor* of the

monthly-calibrated model decreased to 0.75 during the monsoon period, which was compensated during the non-monsoon period (*P-factor*: 1.0).

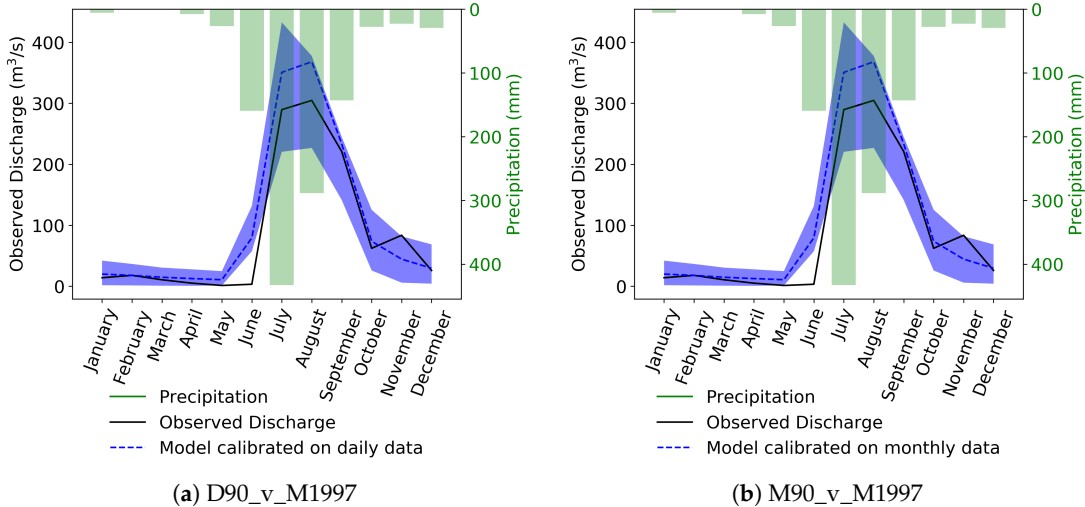

(**a**) D90_v_M1997                                                    (**b**) M90_v_M1997

**Figure 4.** Time-series comparison of SWAT simulated and observed basin outlet monthly discharges for the year 1997, developed using calibrated parameter sets of the calibration cases (**a**) D90 (D90_v_M1997) and (**b**) M90 (M90_v_M1997). The blue band indicates the 95PPU range of simulated discharge; the dotted line indicates the simulation corresponding to the 'best' parameter set; and observations are represented by the solid line.

### 3.3. Further Discussion

The results presented in this study were based on simulations performed on a 5495-km$^2$ agricultural basin with a monsoonal climate. The generalization of these results may be affected by the size, LULC, and hydrological seasonality of the basin.

The generalization of the results across the catchment sizes can be based on the ratios that exist between temporal and spatial scales in hydrological processes, the characteristic velocity [62–64]. Since hydrological scales in length and time tend to be associated and vary with each other [64], these results may be even more pertinent for catchments smaller than the Punpun catchment, which may have quicker hydrological response times (shorter times of concentrations). Hence, information from lower resolution observations (such as monthly discharge) may be more diffused than what is required to calibrate model parameters representatively. For larger basins, it is possible that lower temporal resolution observations may still be acceptable. This is corroborated by studies that model catchments larger than the Punpun basin. For example, SWAT has been used to model large catchments using monthly discharge data for calibration across different spatial resolutions for input (sub-basin level) data: larger than 5000 km$^2$ [14], finer than 100 km$^2$ [13], and for comparisons across different spatial discretizations [65].

The LULC influences rainfall-runoff and hence may impact the generalizability of the results obtained from a primarily agricultural catchment modeled in this study. However, the SWAT model has been seen to perform well across varied conditions (including different LULCs). Moreover, different LULCs, like other input data, will only translate into different parameter sets [66], which are representative of the hydrological processes occurring in the catchment. These parameter sets influence the hydrological response times due to changes in the characteristic velocity, in a manner similar to the influence of catchment size on the response times. For example, forested catchments exhibit a slower response as compared to urban catchments, with agricultural catchments being characterized by intermediate response times [67]. Since the results presented in this study are based on an agricultural catchment with an intermediate response time, calibration of catchments with other LULCs may require observations whose temporal resolutions correspond to their respective hydrological response times.

This implies that for effective calibration, urban catchments may require finer temporal observations than forested catchments.

The hydrological seasonality also needs to be considered before generalizing these results, obtained from a monsoonal region (receiving the majority of the annual rainfall in four months) to other hydro-climatic regions. The results showed that the performance of a daily-calibrated model was much better during the monsoon periods vis-à-vis the non-monsoon period (Section 3.2.4). This is expected as the NSE indicator, used as the objective function during calibration, is biased towards larger differences between the magnitudes of the simulated and observed discharge [68], which occurs more frequently during monsoonal periods. If the model was calibrated with an objective function that assigned similar weightages to smaller and larger discharge values (occurring during non-monsoon and monsoon periods, respectively), the performance of the daily-calibrated models relative to the monthly-calibrated models may have improved even further as the objective function, $NSE$, would not be biased by the larger discharge values. Likewise, for basins with a more uniform distribution of rainfall (and hence discharge) during the year, the same results would hold because of the objective function not being biased by larger discharge values. Hence, hydrological seasonality may not be a factor particularly for basins with more uniform annual distributions of rainfall (such as humid continental or subtropical climates). Since the observed discharge would be expected to have lower variances in such regions compared to monsoonal hydro-climatic regions, a more contrasting difference between the performances of the daily and monthly models could be observed.

Though these issues of size, LULC, and seasonality of a basin are certainly essential to decide the calibration time step for a hydrological model, the inferences drawn in this study may still be applicable. However, more exhaustive modeling studies are required to further generalize these results for wider application by the hydrological modeling community. One suggestion is to employ the newer version of SWAT, SWAT+ [69], for such studies, owing to its improved representation of spatial elements and hydrological processes within the watershed.

## 4. Conclusions

Multiple SWAT models were setup for the Punpun River Basin in India and were calibrated and validated on observed discharge data at two time-steps, daily and monthly, and two different decades, 1980s and 1990s. The monthly-calibrated models of both decades performed slightly better as compared to daily-calibrated models when validated on monthly data, in terms of $P$-$factor$, $R$-$factor$, $R^2$, and $NSE$ indicators. When the models were validated on daily discharge data, it was observed that the monthly-calibrated models performed poorly as compared to daily-calibrated models. This behavior was consistent across models calibrated for both decades and validated on either decade.

Comparison of the calibrated parameters showed that the daily- and monthly-calibrated models had significantly different values for parameters governing channel hydraulic conductivity (CH_K2) and time-lag for soil-water to enter the shallow aquifers (GW_DELAY). These parameters affected the peak runoff rate and quick-flow and base-flow rates, respectively. Consequently, the hydrographs simulated by daily- and monthly-calibrated models had substantial differences. Analysis of simulated daily and monthly streamflow suggested that the models calibrated on monthly data produced unrealistic simulations of daily discharge. Thus, though the monthly-calibrated model captured the patterns in monthly discharge data fairly well, it did not reliably represent daily rainfall-runoff processes in the basin. However, since SWAT is not completely a physically-based hydrological model, a more comprehensive comment on the effect of the temporal resolution of calibration data on model performance requires additional research with purely physically-based models.

The SWAT-like hydrological models (which include model input data and a calibrated parameter set) are frequently used to address problems beyond the range of their current calibration and validation (for instance, to explore cause-effect-based scenarios, or perform 'what-if' analyses, or assess the impacts of climate change). Often, the models developed to address these problems have different calibration and computation time-steps. The results obtained in this study challenge this prevalent

practice of model calibration and recommend that the two time-steps should be the same irrespective of the simulation time-step required for the modeling.

**Author Contributions:** Conceptualization, S.A. and S.T.; Data curation, S.A.; Formal analysis, S.A.; Funding acquisition, S.T.; Investigation, S.A., S.T. and M.D.; Methodology, S.A.; Project administration, S.T.; Resources, S.A.; Software, S.A.; Supervision, S.T. and M.D.; Validation, S.A.; Visualization, S.A.; Writing—original draft, S.A. and S.T.; Writing—review & editing, S.A., S.T. and M.D.

**Funding:** This research was funded by the Ministry of Human Resource Development, Government of India.

**Acknowledgments:** The authors wish to thank James Almendinger and Karim C. Abbaspour for their comprehensive, reliable, and timely responses to SWAT-related queries. The authors also wish to acknowledge the online SWAT and SWAT-CUP user community as a compendium of knowledge related to the SWAT modeling framework and skill set. Finally, the authors thank the Central Water Commission (Government of India) for the provision of discharge data, which was used for model calibration.

**Conflicts of Interest:** The authors declare no conflict of interest. The funders had no role in the design of the study; in the collection, analyses, or interpretation of data; in the writing of the manuscript; nor in the decision to publish the results.

## Abbreviations

The following abbreviations are used in this manuscript:

| | |
|---|---|
| ARS | Agricultural Research Service |
| ASL | Above Sea Level |
| CREAMS | Chemicals, Runoff and Erosion from Agricultural Management Systems |
| CUP | Calibration and Uncertainty Programs |
| CWC | Central Water Commission |
| EPIC | Environmental Impact Policy Climate |
| ET | Evapotranspiration |
| GIS | Geographic Information System |
| GLEAMS | Groundwater Loading Effects on Agricultural Management Systems |
| GLUE | Generalized Likelihood Uncertainty Estimation |
| HRU | Hydrologic Response Units |
| ISRO | Indian Space Research Organization |
| LH | Latin Hypercube |
| LULC | Land Use Land Cover |
| NBSS | National Bureau of Soil Survey |
| NCEP | National Centers for Environmental Prediction |
| NRCS | Natural Resources Conservation Service |
| NRSC | National Remote Sensing Centre |
| NSE | Nash-Sutcliffe Efficiency |
| OAT | One-At-a-Time |
| PPU | Percentage Prediction Uncertainty |
| $R^2$ | Coefficient of determination |
| SCS | Soil Conservation Service |
| SRTM | Shuttle Radar Tomography Mission |
| SUFI2 | Sequential Uncertainty Fitting procedure version 2 |
| SWAT | Soil and Water Assessment Tool |
| USDA | United States Department of Agriculture |

## Appendix A. Results of Global Sensitivity Analysis

**Table A1.** Results of global sensitivity analysis for parameters corresponding to 1980s daily, 1990s daily, 1980s monthly, and 1990s monthly calibration and validation cases (based on Table 3 and Section 2.4.3). The parameter rank, *t*-statistic (*t-stat*) value, and a *p*-value are indicated for each case. The sensitivities are estimates of the change in the objective function due to changes in each parameter (based on a linear approximation), while all other parameters are changing [47]. A larger absolute value of the *t*-statistic (*t-stat*) and a lower *p*-value occur for parameters that are significantly different from 0, subsequently resulting in the relative parameter ranking (a lower numerical rank value is more significant).

| | **1980s Daily** | | | | | | | | | | | |
| | Calibration | | | Validation Cases | | | | | | | | |
| | | | | Self-validation | | | Different time-step, same decade | | | Same time-step, different decade | | |
| | D80_c (1981–1984) | | | D80_v_D80 (1985–1988) | | | D80_v_M80 (1985–1988) | | | D80_v_D90 | | |
| Parameter | Rank | *t-stat* | *p*-value | Rank | *t-stat* | *p*-value | Rank | *t-stat* | *p*-value | Rank | *t-stat* | *p*-value |
|---|---|---|---|---|---|---|---|---|---|---|---|---|
| BASEFLOW ALPHA-FACTOR | 4 | −5.09 | 0 | 4 | −9.47 | 0 | 5 | −3 | 0 | 3 | −10.54 | 0 |
| CH_K(2) | 2 | 10.21 | 0 | 2 | 16.31 | 0 | 4 | 6 | 0 | 2 | 26.35 | 0 |
| CN2 | 7 | −0.49 | 0.62 | 8 | −0.15 | 0.88 | 8 | −0.07 | 0.94 | 7 | 0.35 | 0.73 |
| ESCO | 5 | −1.27 | 0.2 | 5 | −1.92 | 0.05 | 6 | −2.54 | 0.01 | 5 | −2.32 | 0.02 |
| GW_DELAY | 8 | −0.47 | 0.64 | 6 | −1.08 | 0.28 | 3 | −6.36 | 0 | 6 | −2.17 | 0.03 |
| GW_REVAP | 6 | −1.26 | 0.21 | 7 | −0.94 | 0.35 | 7 | 0.61 | 0.54 | 8 | −0.26 | 0.79 |
| SOL_AWC | 3 | 8.14 | 0 | 3 | 12.29 | 0 | 2 | 30.83 | 0 | 4 | 3.39 | 0 |
| SURLAG | 1 | −62.85 | 0 | 1 | −128.97 | 0 | 1 | −39.29 | 0 | 1 | −77.94 | 0 |
| | **1990s Daily** | | | | | | | | | | | |
| | Calibration | | | Validation Cases | | | | | | | | |
| | | | | Self-validation | | | Different time-step, same decade | | | Same time-step, different decade | | |
| | D90_c (1992–1994) | | | D90_v_D90 (1995–1997) | | | D90_v_M90 (1995–1997) | | | D90_v_D80 | | |
| Parameter | Rank | *t-stat* | *p*-value | Rank | *t-stat* | *p*-value | Rank | *t-stat* | *p*-value | Rank | *t-stat* | *p*-value |
| BASEFLOW ALPHA-FACTOR | 3 | −9.56 | 0 | 4 | −8.92 | 0 | 5 | −2.31 | 0.02 | 4 | −9.23 | 0 |

**Table A1.** *Cont.*

| | | | | | | | | | | | | | |
|---|---|---|---|---|---|---|---|---|---|---|---|---|---|
| CH_K(2) | 2 | 24.74 | 0 | 2 | 19.98 | 0 | 4 | 4.03 | 0 | 2 | 19.36 | 0 |
| CN2 | 7 | 0.62 | 0.53 | 7 | 0.82 | 0.41 | 8 | 0.05 | 0.96 | 7 | 0.68 | 0.49 |
| ESCO | 5 | −5.33 | 0 | 6 | −3.26 | 0 | 7 | −0.39 | 0.7 | 5 | −3.51 | 0 |
| GW_DELAY | 6 | −1.98 | 0.05 | 5 | −3.34 | 0 | 1 | −20.88 | 0 | 6 | −1.37 | 0.17 |
| GW_REVAP | 8 | −0.57 | 0.57 | 8 | 0.1 | 0.92 | 6 | −1.61 | 0.11 | 8 | −0.21 | 0.83 |
| SOL_AWC | 4 | 8.94 | 0 | 3 | 10.54 | 0 | 2 | 12.59 | 0 | 3 | 9.92 | 0 |
| SURLAG | 1 | −95.1 | 0 | 1 | −93.61 | 0 | 3 | −10.56 | 0 | 1 | −89.9 | 0 |

**1980s Monthly**

| | Calibration | | | Validation Cases | | | | | | | | |
|---|---|---|---|---|---|---|---|---|---|---|---|---|
| | | | | Self-validation | | | Different time-step, same decade | | | Same time-step, different decade | | |
| | M80_c (1981–1984) | | | M80_v_M80 (1985–1988) | | | M80_v_D80 (1985–1988) | | | M80_v_M90 | | |
| Parameter | Rank | *t-stat* | *p*-value | Rank | *t-stat* | *p*-value | Rank | *t-stat* | *p*-value | Rank | *t-stat* | *p*-value |
| CH_K(2) | 1 | −40.2 | 0 | 1 | 12.41 | 0 | 1 | 117.64 | 0 | 2 | −11.24 | 0 |
| CN2 | 6 | 1.26 | 0.21 | 5 | 1.06 | 0.29 | 4 | 1.72 | 0.09 | 1 | −22.84 | 0 |
| ESCO | 4 | 2.43 | 0.02 | 4 | −1.15 | 0.25 | 3 | −2.74 | 0.01 | 3 | −5.74 | 0 |
| GW_DELAY | 2 | −22.33 | 0 | 2 | 11.58 | 0 | 5 | −0.85 | 0.4 | 4 | −2.03 | 0.04 |
| GW_REVAP | 5 | −1.85 | 0.07 | 6 | 0.04 | 0.97 | 6 | −0.76 | 0.44 | 5 | 0.81 | 0.42 |
| SOL_AWC | 3 | −13.48 | 0 | 3 | 5.29 | 0 | 2 | 15.3 | 0 | 6 | 0.78 | 0.43 |

**1990s Monthly**

| | Calibration | | | Validation Cases | | | | | | | | |
|---|---|---|---|---|---|---|---|---|---|---|---|---|
| | | | | Self-validation | | | Different time-step, same decade | | | Same time-step, different decade | | |
| | M90_c (1992–1994) | | | M90_v_M90 (1995–197) | | | M90_v_D90 (1995–197) | | | M90_v_M80 | | |
| Parameter | Rank | *t-stat* | *p*-value | Rank | *t-stat* | *p*-value | Rank | *t-stat* | *p*-value | Rank | *t-stat* | *p*-value |
| CH_K(2) | 1 | −35.51 | 0 | 1 | −14.39 | 0 | 1 | 112.91 | 0 | 2 | 9.44 | 0 |
| CN2 | 4 | 1.04 | 0.3 | 5 | −0.48 | 0.63 | 6 | −0.27 | 0.78 | 4 | −1.15 | 0.25 |
| ESCO | 6 | 0.2 | 0.84 | 6 | −0.39 | 0.7 | 4 | −2.51 | 0.01 | 6 | −0.72 | 0.47 |
| GW_DELAY | 2 | −11.04 | 0 | 2 | −12.93 | 0 | 2 | −3.61 | 0 | 1 | 12.04 | 0 |
| GW_REVAP | 5 | −0.41 | 0.68 | 4 | −1.19 | 0.24 | 5 | −1.07 | 0.28 | 3 | 2.18 | 0.03 |
| SOL_AWC | 3 | −3.08 | 0 | 3 | −1.65 | 0.1 | 3 | 3.05 | 0 | 5 | 1.06 | 0.29 |

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
