# Peer review of "Can We Calibrate a Daily Time-Step Hydrological Model Using Monthly Time-Step Discharge Data?"

_water, doi:10.3390/w11091750_

Round 1

Reviewer 1 Report

I find the paper very interesting for the modelling community and very well written and structured. The topic addressed by the authors is important since many times we have to rely on larger time-step calibration to accept results of finer temporal resolution and the question is if this can be acceptable. The methodology of the paper to address this question is correct and adequate since the authors test their monthly and daily data and calibration time-steps at two different periods and they use four different statistics to evaluate the results. All their modelling work seems well performed and I have not any concerns related to it. The results and conclusions drawn are well supported by the outputs of the model. My main suggestion for this article would be to have a clearer and longer discussion part. I have only a few comments which I list them below:

1) Although a combined 'Results and Discussion'section is appropriate (personally I prefer it in research papers), there are issues which could be summarized in a separate 'Discussion' section. For example, the size of the watershed used as case study for this epxeriment could play a role on the results. What about having a very small or very large (Region) area for modelling. Could a monthly calibration be more acceptable in a daily flow study when the sale is very small (small times of concentrations) or even less acceptable? Can the authors elaborate on this? The paper could give a useful message if discussed such issues and its status could increase if it becomes more generic. I understand that applying the method to various watersheds is not feasible for an individual paper but some discussion would be useful. 

2) Another paper worthy for citation in the introduction is a calibration paper at the large scale where a lot of calibration issues are discussed (including also water quality, which is not addressed in the present paper). See here: https://doi.org/10.1016/j.jhydrol.2015.02.039.

3) The ESCO parameter in the newest SWAT versions can be adjusted at the HRU scale through the 'HRU'files. Can the authors check and modify their text accordingly at the end of page 8.Just to be accurate whether SWAT differentiates the ESCO parameter at the basin scale only or not. 

Author Response

Response to Reviewer 1 Comments

Point 1: Although a combined 'Results and Discussion' section is appropriate (personally I prefer it in research papers), there are issues which could be summarized in a separate 'Discussion' section. For example, the size of the watershed used as case study for this experiment could play a role on the results. What about having a very small or very large (Region) area for modelling. Could a monthly calibration be more acceptable in a daily flow study when the sale is very small (small times of concentrations) or even less acceptable? Can the authors elaborate on this? The paper could give a useful message if discussed such issues and its status could increase if it becomes more generic. I understand that applying the method to various watersheds is not feasible for an individual paper but some discussion would be useful. 

Response 1: Additional sub-section “Further Discussion” is inserted to discuss the role of size, seasonality and LULC of the basin on the choice of calibration time step. 

Point 2: Another paper worthy for citation in the introduction is a calibration paper at the large scale where a lot of calibration issues are discussed (including also water quality, which is not addressed in the present paper). See here: https://doi.org/10.1016/j.jhydrol.2015.02.039.

Response 2: We have cited this reference in the section “Introduction” (Line number 50) and the sub-section “Further discussion” (Line number 451).

Point 3: The ESCO parameter in the newest SWAT versions can be adjusted at the HRU scale through the 'HRU' files. Can the authors check and modify their text accordingly at the end of page 8. Just to be accurate whether SWAT differentiates the ESCO parameter at the basin scale only or not. 

Response 3: The ESCO parameter is modifiable at both basin and HRU levels, as it is present in the *.bsn file as well as the multiple *.hru files (https://swat.tamu.edu/media/69296/swat-io-documentation-2012.pdf, pages 95 and 237). It is also mentioned as a basin scale parameter in other peer-reviewed publications (10.1016/j.jhydrol.2018.10.024, Table A1, pg 680). However, a modification has been made in the specified line to include the suggestion as well, as follows: “The parameters ‘ESCO’ and ‘SURLAG’ were calibrated at the basin scale. While SURLAG is defined as a single basin-wide value, SWAT allows ESCO to be modified at both the basin as well as HRU scale.”

Reviewer 2 Report

I find this work both interesting and credible. My only concern relates to Figure 3 iii. Here the model does not seem to match the measurements well. Some comment on this in the caption would seem appropriate. Moreover, the authors' decision to present an x-axis that jumps from May to October is somewhat confusing. Following clarification of this issue, I would recommend publication.

Author Response

Response to Reviewer 2 Comments

Point 1: I find this work both interesting and credible. My only concern relates to Figure 3 iii. Here the model does not seem to match the measurements well. Some comment on this in the caption would seem appropriate. Moreover, the authors' decision to present an x-axis that jumps from May to October is somewhat confusing. Following clarification of this issue, I would recommend publication.

Response 1: Extra explanatory line added to the list figure caption (to illustrate the non-performance of both models in October-November 1997, Figure 3 iii). It is reproduced here: “Note here that both the daily and monthly calibrated models, with their best simulations and respective 95PPU bands, are unable to capture the large discharge in the hydrograph from 26th October to 12th November 1997.”

Axis jump has also been worked on - instead of a jump, there are two figures which indicate the two separate non-monsoonal periods within the calendar year (January-May, October-December).